# HIV preventive practice and its associated factors among street dwellers in Ethiopia: Application of health belief model

Yosef Wassihun[1], Zemed Hunegnaw[2], Tadele Fentabel Anagaw [1], Zeamanuel Anteneh Yigzaw[1], Eyob Ketema Bogale [1] *

1 Health Promotion and Behavioral Sciences Department, Bahir Dar University, Bahir Dar, Amhara Region, Ethiopia, 2 Health Promotion and Behavioral Sciences Department, Amhara Regional Health Bureau, Bahir Dar City, Amhara Region, Ethiopia

☯ These authors contributed equally to this work.
* ketema.eyob@gmail.com

**Data Availability Statement:** All data are in the manuscript and /or supporting information files.

**Funding:** The authors received no specific funding for this work.

## Abstract

Street dwelling is the use of public space as a place of residence and nourishment for children, adolescents, and young people, revealing a complex and multifactorial situation. Moreover, homelessness facilitates HIV transmission and its progression due to the risky lifestyle of individuals. To the best of our knowledge, there is no study conducted in the study area on HIV preventive practices among street dwellers. The aim of this study was to assess HIV prevention practices and associated factors among street dwellers. A community-based cross-sectional study was conducted in Bahir Dar City from March 12, 2023 to April 30, 2023. By using the simple random sampling technique, 424 street dwellers were recruited. Bivariable and multivariable logistic regression were used for analysis. The magnitude of HIV preventive practice among street dwellers was 35.9%. Being male (AOR = 0.23, 95% CI: 0.10, 0.55), educational status (AOR = 7.53, 95% CI: 2.20, 25.6), practice of sex to earn money (AOR = 0.18, 95% CI: 0.08, 0.44), good knowledge about HIV preventive practice (AOR = 2.83, 95% CI: 1.46, 5.49), perceived susceptibility for HIV (AOR = 0.90, 95% CI: 0.81, 0.99), and perceived benefit of using HIV preventive practice (AOR = 1.09, 95% CI: 1.02, 1.17), were factors associated with HIV preventive practice. The magnitude of HIV preventive practice was low. Being male, the ability to read and write, the practice of sex to earn money, good knowledge about HIV preventive practice, perceived susceptibility to HIV, and the perceived benefit of using HIV preventive practice were significantly associated with HIV preventive practice. Therefore, responsible organizations, both governmental and non-governmental, should design inclusive strategies to improve HIV preventive practice among street dwellers by focusing on regular demand creation activities, awareness creation about HIV preventive practice, and sustainable condom distribution in the city.

**Competing interests:** The authors have declared that no competing interests exist.

**Abbreviations:** AIDS, Acquired Immune Deficiency Syndrome; ANRS, Amhara National Regional State; AOR, Adjusted Odds ratio; BDU, Bahir Dar University; CI, Confidence Interval; COR, Crude Odds Ratio; CSW, Commercial Sex Workers; HBM, Health Belief Model; HIV, Human Immune Deficiency Virus; EDHS, Ethiopian Demographic and Health Survey; HAPCO, HIV prevention and Control Office; HH, House Hold; LMIC, Low- and Middle-Income Countries; PI, Principal Investigator; SDG, Sustainable Development Goal; SPSS, Statistical Package for Social Science Studies; STI, Sexually Transmitted Infections; UNAIDS, United Nations Program on AIDS; VCT, Voluntary Counseling and Blood Testing; WHO, World Health Organization.

## Introduction

HIV prevention practices like abstinence, delaying sexual initiation, faithfulness, fewer sexual partners, routine condom use, avoiding commercial sex, refraining from injecting drugs, combating violence against women, and being aware of one's HIV status are promoted through information, education, and communication [1].

Worldwide, around 28 million street dwellers live on the street, and the HIV/AIDS epidemic and other difficulties affecting parents and guardians are linked to the dramatic rise in the number of homeless people [2]. HIV/AIDS prevalence in Iran among street dwellers was reported at 3.4% [2].

HIV/AIDS transmission is highest among those living on the streets in the United States [3]. It appears that heightened rates of HIV/AIDS risk behaviors contribute to the apparent explanation of the disproportionate burden of HIV among street people or people living in unstable housing. In studies conducted in the U.S., there were associations between street dwellers and HIV/AIDS risk behaviors, such as an increased likelihood of having multiple sexual partners [4].

The relationship between housing and HIV/AIDS risk and health outcomes has been acknowledged in policies, including in the United States (U.S.). The National HIV/AIDS Strategy, which calls for efforts to reduce street dwellers and enhance housing stability as part of a comprehensive approach to stop street people from engaging in high-risk behaviors, is necessary to avoid long-term negative effects on the health system and society [5]. Street life is full of problems due to the difficulties of living on the street, where there is little access to food, money, or sleeping spaces. Some are engaged in transactional sexual behavior, the practice of homosexuality, dependency on alcohol, and addiction to drugs to make ends meet, but in doing so, they also increase their risk of contracting HIV [6]. HIV/AIDS among street-connected dwellers is a big issue, especially in countries with lower incomes, and they have been left participating in extremely dangerous activities that put them at risk for HIV infection due to poor protection and care, a lack of understanding of the risk of contracting HIV, and a lack of or insufficient access to health services for people connected to the street [7].

Because of financial difficulties and previous life circumstances, the number of street dwellers in Ethiopian cities is significantly rising, particularly in major cities [8, 9]. According to a recent report from Ethiopia's Ministry of Labour and Social Affairs, there were about 24,000 homeless people in Addis Ababa in 2018, including about 10,500 homeless kids and 13,500 homeless adults [10]. Most street dwellers are at high risk of many risky behaviors, including having multiple partners for sex at once, early introduction of sex, engaging in unprotected sex, group sex, using shared needles while abusing drugs, and working in sex [11].

Despite the rise in the number of homeless people, HIV preventive practices and programs for street dwellers are unable to meet their specific reproductive and sexual health needs [8]. The circumstances in which street dwellers live increase their vulnerability to HIV/AIDS [11]. Currently, besides using the abstain, be faithful, and consistent condom use (ABC) model to prevent HIV AIDS, voluntary medical male circumcision [12] and pre-exposure prophylaxis (PrEP) [13] have emerged as human immunodeficiency virus (HIV) prevention tools for populations at highest risk for HIV infection, but they have not been considered in our study since they are not much known among street dwellers as the preventive practice of HIV.

An attempt is made to explain and forecast health behavior using the Health Belief model (HBM), a psychological model. This is accomplished through concentrating on people's attitudes and beliefs. According to the model, the constructs of perceived vulnerability, perceived severity, perceived advantage, perceived barriers, cues to action, and self-efficacy determine behavior. Therefore, the model helps to identify individuals' levels of risk, the consequences of

certain health problems, and the use of the recommended activity to avert the health problem. It also helps to understand individuals' level of confidence to take the recommended action and their strategic readiness for the accomplishment of the recommended behaviour [14].

Besides using the HBM framework, this study was conducted at Bahir Dar city, since it was a promising city for employment, the state capital, a well-liked tourist attraction, and a prominent religious center, it had a high proportion of street dwellers moving there from rural areas [15]. The reason that motivates me to study the HIV preventive practice of street dwellers is that, to the best of our understanding, no study has been done in the study area on the HIV prevention practices among street dwellers, and from our personal experience, we observed many difficulties amongst these demographic groups.

Therefore, this study aims to assess HIV prevention practices and associated factors among street dwellers through the application of HBM to fill the gaps.

## Methods and materials

### Study setting and period

The study was conducted in Bahir Dar City from March 12 to April 30, 2023. Bahir Dar is one of the most popular tourist sites in Ethiopia, and the area around Lake Tana, the source of the Blue Nile River, is home to many attractions. Six sub-cities and three public hospitals are located inside the city. The city is filled with markets, structures, asphalt, cobblestone streets, churches, and mosques. These structures and places of worship also help those living on the streets survive outside of their main function. According to the Bahir Dar Women and Child Affairs Office report, there are 1231 street dwellers available in this city, of whom 758 are above the age of 12. The research was carried out between March 12, 2023, and April 30, 2023.

### Study design

A community-based cross-sectional study design was conducted.

### Population

**Source of population.**   All street dwellers living in Bahir Dar City.

### Study population

All street dwellers living in Bahir Dar City during the data collection period.

### Inclusion criteria

All street dwellers residing in Bahir Dar City whose ages are greater than or equal to 12 years old were included.

### Sample size determination

The sample size (n) required for the study was calculated using a single population proportion formula by considering the following assumptions: $Z\alpha/2$ = the critical value for a normal distribution at 95% CI is 1.96, 5% margin of error, and P = 50% (since there is no published research on HIV preventive practice and its associated factors among street dwellers in Ethiopia). The final sample size was 424 after adding a 10% non-response rate.

### Study variable

**Dependent variable.**   HIV preventive practice (yes, no).

## Independent variables

- **Socio-demographic characteristics:** age, sex, marital status, educational status, religion, and source of income

- Knowledge

- Attitude

- Substance use

- **Constructs of HBM:** perceived susceptibility, perceived severity, perceived benefit, perceived barriers for HIV prevention, self-efficacy, and cues to action.

## Operational definition and measurement

**Street dwellers are** those who live and work on the street.

**HIV preventive practice** was measured by three items. Participants who used at least one of the three HIV preventive practices (1, abstaining from sexual intercourse), (2) having only one sexual partner (being faithful), and (3) persistent condom use were classified as having preventive practice toward HIV, whereas those who did not use at least one preventive practice were classified as not having preventive practice towards HIV [16].

**Knowledge:** It was measured by using five yes-or-no questions. These questions were summed up with a 1 minimum and a 5 maximum score. The mean value was calculated, and participants who scored greater than the mean score were considered to have good knowledge and others to have poor knowledge [17].

**Substance use:** It was measured by nine "yes/no" questions. The participants who took at least one type of substance were considered substance users, whereas those who did not take at least one type of substance were considered non-substance users [18].

**Attitude:** Four items with five-point Likert scale questions were used to measure it, and the sum of these items produced a total score that ranged from 4 to 20. Participants who scored above the mean value were classified as having a favorable attitude, and those who scored below the mean value were classified as having an unfavorable attitude [19].

**Perceived susceptibility: I**t is a conviction regarding one's propensity to contracting a disease or other ailment. It consists of five items on a Likert scale with a maximum score of 25, with scores for each item ranging from a minimum of 5 to a maximum of 25. Perceived severity consists of three items on a five-point Likert scale, and the score of individuals for each item was summed up and ranged from 3 minimums to 15 maximum total scores. It is a Belief about the seriousness of contracting a disease or condition, including its consequences.

**Perceived benefit** consists of eight items on a five-point Likert scale, and the score of individuals for each item was summed up and ranged from an 8 minimum to a 40 maximum total score. It is Belief in the positive aspects of adopting healthy behavior. **Perceived barriers:** It is a belief about obstacles to performing a behavior. It consists of ten items on a Likert scale. The scores of respondents for each item were added up and varied from 10 to 50. **Self-efficacy:** It is the Belief that one can perform the recommended health behavior (confidence). It has nine items on a Likert scale with a maximum score of 45 and a minimum score of 9, and each item has a Likert scale with five possible responses. **Cues to action:** it is Internal or external factors could trigger health behaviors. It was composed of ten items on a Likert scale with a maximum score of 50 and a minimum score of 10. Each item had a five-point Likert scale. Individuals' scores for each item were added together for each construct, and all constructs were considered as continuous variables for analysis [20].

## Data collection tools, method and sampling procedure

A structured, interviewer-administered questionnaire (S1 Text) was used to collect the data. The questionnaire asked about sociodemographic factors, knowledge, drug use, HIV prevention practices, attitude, and perception of HIV prevention practices (including perceived susceptibility, perceived severity, perceived benefit, perceived barriers, self-efficacy, and cues to action). The questionnaires were adapted from similar studies [21–24]. The desired sample was selected by simple random sampling using the lottery method.

## Data quality control

To ensure consistency, the questionnaire was translated from English to Amharic and then back to English by an independent person. A comparable population in the research region of Finote Selam town served as the questionnaire's pre-testing group. Six data collectors (BSc holders in public health) and one supervisor (an MPH holder) were recruited and trained for three days before the data collection. The collected data were evaluated for completeness, clarity, and consistency by the supervisor and principal investigator daily. The data collection process was challenging due to the nature of street dwellers' mobility. We used nighttime and early morning to get the study participants before they went away from their temporal stations. Also, street dwellers' fingers were painted with permanent marker to avoid double counting during data collection.

A person who is proficient in both languages translated the questionnaire from English into Amharic and back again. A pre-test was done on 22 street dwellers in Finote Selam town. One day of training was given to six data collectors and one supervisor. Reliability was checked using Cronbach's' alpha > 0.7, which was >0.9.

## Data processing and analysis

Using Epi-data software version 3.1, the data were verified, coded, and entered before being exported to SPSS version 25 for additional analysis. For all of the variables, frequency and proportions were used to show the descriptive result. In a bivariate analysis, each independent variable was evaluated for a statistically significant relationship with the dependent variable at a 95% confidence level and a p-value of less than 0.25. The final multiple logistic regression model included the variables that had p-values less than 0.25 in the bivariate analysis to account for potential confounders. A p-value of 0.05 or less was deemed statistically significant in the final model. The Hosmer and Lemeshow test of best fit, which yielded a p-value of >0.05 and 0.709, was used to assess the final model's goodness of fit.

## Ethical considerations

This study was conducted according to the guidelines laid down in the Declaration of Helsinki, and ethical approval was obtained from the Institutional Review Board of Bahir Dar University with reference number 727/2023. Also, an official letter was obtained from the Amhara Public Health Institute. Written informed consent was obtained from each study participant after informing them of the purpose, benefits, and risks.

For participants<18 years old, the Bahir Dar City Women and Child Affairs Office was asked to provide consent for their participation, approval, and consent to participate. Permission to undertake the study was obtained at all levels. Bahir Dar city women and child affairs office were given detailed information about the purpose of the study, data collection procedures, and possible risks, discomforts, and benefits of participating in the study through the consent process. Written informed consent was obtained from the Bahir Dar city women and

child affairs office. Despite the Bahir Dar city women and child affairs office's consent, a child's decision not to participate in the study was respected.

Participants who couldn't read or write had their consent forms read to them by the data collector, so they were aware of all of the conditions. Data collection procedures used codes rather than participant names to ensure the confidentiality of the information collected from study participants.

## Results

### Socio-demographic and substance use characteristics of study participants

A total of 424 study participants were involved in the study, with a response rate of 96.7%. Of the total participants, 14 were excluded from the data collection process, and we consider them non-responders. The mean ages of the respondents were 18.01 (SD + 4.8) years. About two hundred sixty-six of the participants were male, and two hundred eighty-six (69.8%) were between the ages of 12 and 19 years. Two hundred eighty-eight (70.2%) of the respondents were orthodox, and 289 (70.5%) of the respondents were single. More than half of the participants (54.9%) had attended primary school. In this study, 384 (93.7%), 287 (70%), 283 (69%), and 52 (12.7%) of street dwellers used alcohol, tobacco, khat, and hashish, respectively (Table 1).

### Knowledge of respondents about HIV preventive practice

In this study, about 239 (58.3%) of the respondents had good knowledge about HIV preventive practices. One hundred eighty-four (44.9%) of the participants responded that HIV can be prevented by the correct use of a consistent condom at every sexual intercourse, and 199 (48.5%) responded that only one faithful sexual partner can prevent the risk of getting HIV/AIDS (Table 2).

### Attitudes of participants about HIV/AIDS preventive practice

In this result, about 259 (63.2%) of participants have a favorable attitude toward HIV/AIDS preventive practice (Fig 1).

### Preventive practice of HIV/AIDS among street dwellers

In this study, the magnitude of HIV preventive practice was 35.9% (95% CI, 31.2, 40.7), and 368 (89.8%) of the participants had sexual intercourse after being street dwellers (Fig 2).

### Perception of street dwellers towards HIV preventive practice

The mean scores of perceived susceptibility, perceived severity, perceived barrier, perceived benefit, self-efficacy, and cues to action were 14 (SD ± 4.41), 13.1 (SD ± 4.44), 19.99 (SD ± 6.58), 27.9 (SD ± 8.59), 19.7 (SD ± 7.01), and 27.1 (SD ± 9.21), respectively (Table 3).

### Factors associated with HIV preventive practices among street dwellers

In the bivariate logistic regression analysis, educational status, the practice of sex to earn money, sex, age, marital status, knowledge, attitude towards HIV/AIDS, perceived benefit, self-efficacy, perceived susceptibility, cues to action and perceived severity were the determinants of HIV preventive practices at a p-value of 0.25, and these variables were included in the multivariable logistic regression analysis. In multivariate logistic regression, sex, educational

**Table 1. Socio-demographic and substance use characteristics of street dwellers in Bahir Dar City, 2023 (n = 410).**

| Variables | Category | Frequency | Percent |
|---|---|---|---|
| Sex (n = 410) | Male | 243 | 59.3 |
| | Female | 167 | 40.7 |
| Age | 12–19 years | 286 | 69.8 |
| | 20–24 years | 74 | 18.0 |
| | 25–34 years | 50 | 12.2 |
| Religion | Orthodox Christian | 288 | 70.2 |
| | Muslim | 98 | 23.9 |
| | Protestant | 24 | 5.90 |
| Marital status | Single | 289 | 70.5 |
| | Divorced | 92 | 22.4 |
| | Widowed | 24 | 5.90 |
| | Married | 5 | 1.20 |
| Educational status | Illiterate | 34 | 8.30 |
| | Read and write | 124 | 30.2 |
| | Primary school(1 to 8 grade) | 225 | 54.9 |
| | Secondary school and above | 27 | 6.60 |
| Work to earn money | Yes | 385 | 93.9 |
| | No | 25 | 6.10 |
| If yes, types of work to earn money | Begging | 148 | 36.1 |
| | Shoe shining | 33 | 8.00 |
| | Carrying | 151 | 36.8 |
| | Practice of sex to earn money | 177 | 43.7 |
| Do you use alcohol? | Yes | 384 | 93.7 |
| | No | 26 | 6.30 |
| Frequency of alcohol use | Weekly | 46 | 12.0 |
| | Two times/week | 183 | 47.7 |
| | Three times/week | 73 | 19.0 |
| | > three times/week | 82 | 21.4 |
| Type of alcohol used | Arekie | 343 | 89.3 |
| | Tella | 248 | 64.6 |
| | Tej | 23 | 6.00 |
| | Beer | 17 | 4.40 |
| Do you use tobacco/ cigarette/? | Yes | 287 | 70.0 |
| | No | 123 | 30.0 |
| Frequency of smoking tobacco/ cigarette | Weekly | 49 | 17.1 |
| | Two times/week | 51 | 17.8 |
| | Three times/week | 78 | 27.2 |
| | > three times/week | 109 | 38.0 |
| Do you use Khat? | Yes | 283 | 69.0 |
| | No | 127 | 31.0 |
| Frequency of Khat chewing | Weekly | 22 | 7.80 |
| | Two times/week | 57 | 20.1 |
| | Three times/week | 100 | 35.3 |
| | > three times/week | 104 | 36.7 |
| Do you use hashish? | Yes | 52 | 12.7 |
| | No | 358 | 87.3 |

(*Continued*)

**Table 1.** (Continued)

| Variables | Category | Frequency | Percent |
|---|---|---|---|
| Frequency of using hashish | Weekly | 22 | 42.3 |
| | Two times/week | 23 | 44.2 |
| | Three times/week | 7 | 13.5 |

**Khat**: the leaves of the shrub Catha edulis which are chewed like tobacco or used to make tea; has the effect of euphoric stimulant

status, the practice of sex to earn money, knowledge, perceived susceptibility, and perceived benefit were significantly associated with HIV preventive practices at a p-value of < 0.05.

Being male were 2.3 times more likely to have HIV preventive practice than female (AOR = 2.3; 95% CI: 1.09, 3.55). The street dwellers who completed elementary and above were 7.53 times more likely to have HIV preventive practices than those who are illiterate (AOR = 7.53, 95% CI: 2.20, 25.6). Those street dwellers who did not practice sex to earn money were 1.52 times more likely to have HIV preventive practices than those who practice sex to earn money (AOR = 1.52, 95% CI:1.04, 6.41). The street dwellers who had good knowledge were about 2.83 times more likely to have HIV preventive practices than those who had poor knowledge (AOR = 2.83, 95% CI: 1.46, 5.49).

With One unit increase in perceived susceptibility towards HIV preventive practice, the odds of HIV preventive practice are increased by 0.9 (AOR = 0.90, 95% CI: 0.81, 0.99). With one unit increase in perceived benefit towards HIV preventive practice, the odds of HIV preventive practice are increased by 1.09 (AOR = 1.09, 95% CI: 1.02, 1.17) (Table 4).

## Discussion

The findings of this study revealed that the magnitude of HIV preventive practice among street dwellers in Bahir Dar City was 35.9% (95% CI: 31.2, 40.7). This finding is in line with studies done in southern Brazil (38.1%) [25], California (39.9%) [26], and Ukraine (39.4%) [27].

This study showed that males are more likely to have HIV prevention practices. This finding is supported by a study done in southern Ethiopia, Hawassa [28]. This might be due to men are more likely to control their sexual pleasure and be able to determine the time and

**Table 2. Knowledge about HIV preventive practice among street dwellers in Bahir Dar City, 2023 (n = 410).**

| Variable | Category | Frequency | Percent |
|---|---|---|---|
| Can HIV be prevented by the correct use of condom at every sexual intercourse? (n = 410) | Yes | 184 | 44.9 |
| | No | 226 | 55.1 |
| Having only one and faithful sexual partner can prevent the risk of getting HIV/AIDS? | Yes | 199 | 48.5 |
| | No | 211 | 51.5 |
| Can HIV be prevented by abstinence? | Yes | 253 | 61.7 |
| | No | 157 | 38.3 |
| Can HIV transmitted by eating together with an HIV infected person? | Yes | 67 | 16.3 |
| | No | 343 | 83.7 |
| May a healthy looking person have the HIV virus in his/her blood? | Yes | 66 | 16.1 |
| | No | 344 | 83.9 |
| Knowledge | Poor Knowledge | 171 | 41.7 |
| | Good Knowledge | 239 | 58.3 |

HIV: Human Immune Deficiency Virus, AIDS: Acquired Immune Deficiency Syndrome

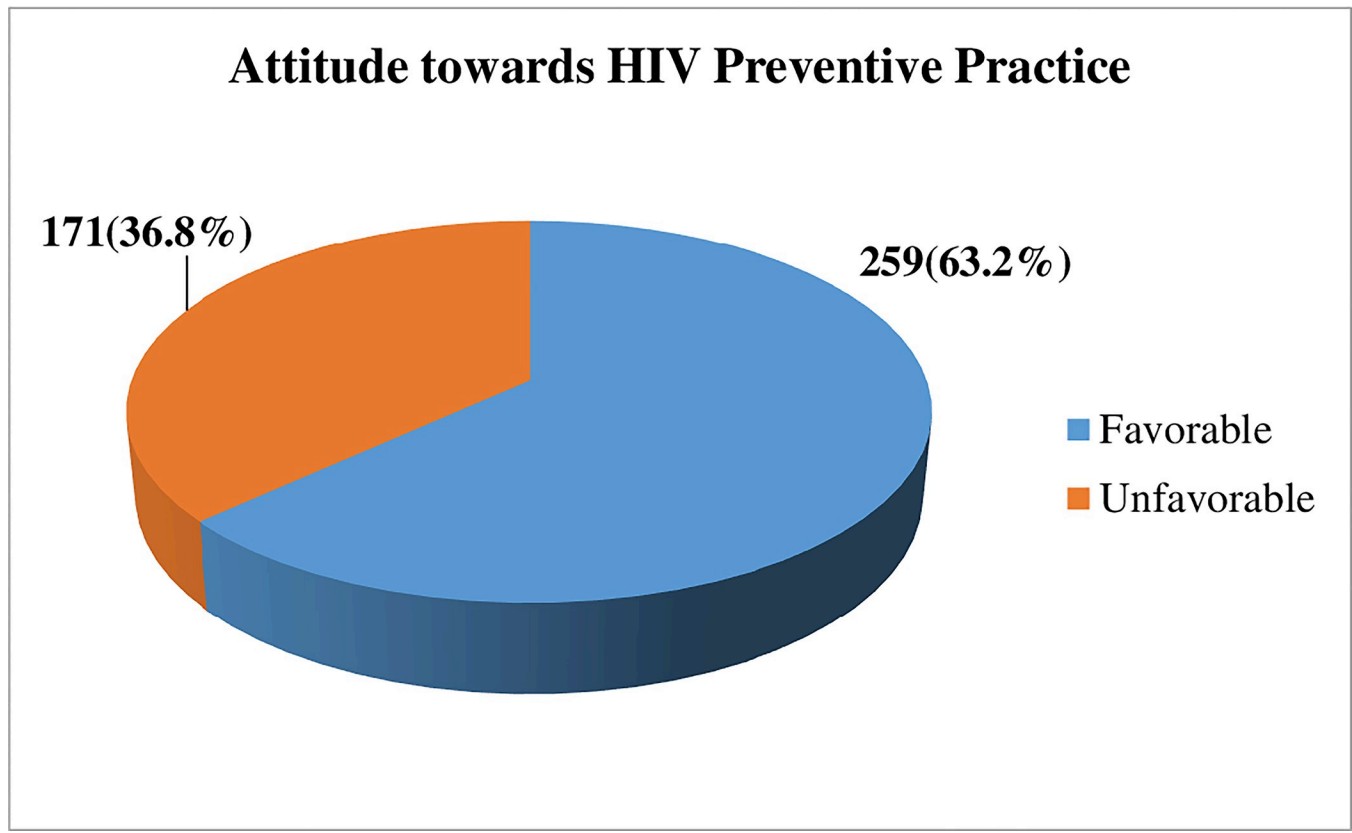

**Fig 1. Attitude towards HIV prevention practice among street dwellers in Bahir Dar City, 2023 (n = 410).**

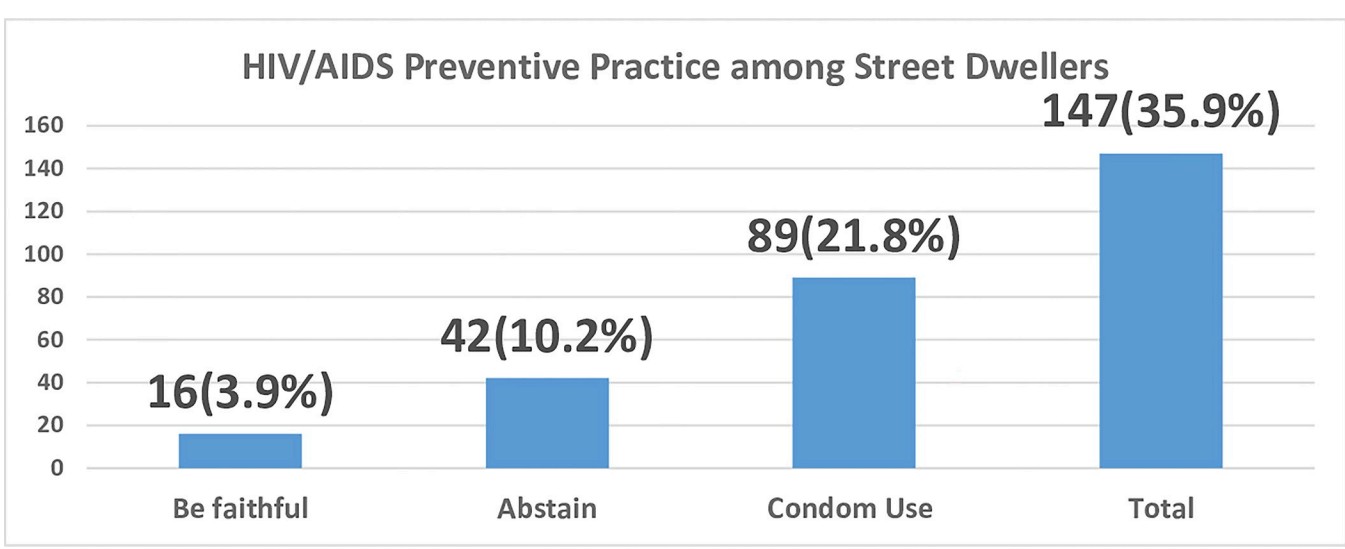

**Fig 2. HIV preventive practice among street dwellers in Bahir Dar City, 2023 (n = 410).**

**Table 3. Descriptive statistics of perception towards HIV preventive practice among street dwellers in Bahir Dar City, 2023 (n = 410).**

| Constructs | No of Items | Minimum value | Maximum value | Mean | SD | Reliability test (Cronbach's' alpha) |
|---|---|---|---|---|---|---|
| Perceived susceptibility | 5 | 5 | 25 | 14 | 4.41 | 0.95 |
| Perceived severity | 4 | 4 | 20 | 13.1 | 4.44 | 0.99 |
| Perceived benefit | 7 | 7 | 35 | 19.9 | 6.68 | 0.96 |
| Perceived barrier | 10 | 10 | 50 | 27.9 | 8.59 | 0.95 |
| Self-efficacy | 9 | 9 | 45 | 19.7 | 7.01 | 0.96 |
| Cues to action | 9 | 9 | 45 | 27.1 | 9.21 | 0.97 |

place where sex takes place than women, which increases the probability of using HIV preventive practices.

The current study showed that participants who had good knowledge had a significant association with HIV preventive practice. This study finding was supported by other study findings in south-west Ethiopia [16], Dima district of the Gambella region [29], Gambella town [21], Majang zone-Gambella region [30], Afar region [31], Jigawa State, Nigeria [32], and Indonesia [33]. This might be due to the fact that good knowledge will produce good behavior, and people who are aware of HIV/AIDS will act responsibly when engaging in sexual activity [33].

**Table 4. Factors associated with HIV preventive practice among street dwellers in Bi-variable and multiple logistic regression analysis in Bahir Dar City, 2023 (n = 410).**

| Variable | Category | HIV preventive practices | | COR (95% CI) | AOR (95% CI) | P-Value |
|---|---|---|---|---|---|---|
| | | Yes | No | | | |
| Sex (n = 410) | Male | 116 | 150 | 2.81(1.86, 4.96) | 2.3 (1.09, 3.55) | 0.00 |
| | female | 31 | 113 | 1.00 | 1.00 | |
| Age | 12–19 years | 108 | 178 | 1.00 | 1.00 | |
| | 20–24 years | 26 | 48 | 0.89(0.52, 1.52) | 1.12(0.54, 2.36) | 0.76 |
| | 25–34 years | 13 | 37 | 0.58(0.29, 1.14) | 1.54(0.57, 4.15) | 0.39 |
| Marital status | Single | 118 | 171 | 1.00 | 1.00 | |
| | Divorced | 20 | 66 | 0.43(0.34, 0.95) | 0.51(0.24, 1.08) | 0.78 |
| | Married and widowed | 9 | 26 | 0.50(0.05, 0.56) | 0.38(0.08, 1.79) | 0.22 |
| Educational status | Unable to R & W | 8 | 26 | 1.00 | 1.00 | |
| | Elementary+ | 55 | 69 | 2.59(1.09, 6.17) | 7.53(2.20, 25.6) | 0.00 |
| | able to read & write | 84 | 168 | 1.63(0.71, 3.74) | 1.75(0.55, 5.55) | 0.34 |
| Practice of sex to earn money | No | 98 | 145 | 1.78 (1.23, 4.74) | 1.52 (1.04, 6.41) | 0.00 |
| | Yes | 39 | 103 | 1.00 | 1.00 | |
| Knowledge about HIV/AIDS | Good | 115 | 124 | 4.03 (2.54, 6.38) | 2.83(1.46, 5.49) | 0.01 |
| | Poor | 32 | 139 | 1.00 | 1.00 | |
| Attitude towards HIV/AIDS | Favorable | 120 | 139 | 3.97(2.45, 6.42) | 1.41 (0.51, 3.85) | 0.51 |
| | Unfavorable | 27 | 124 | 1.00 | 1.00 | |
| Perceived susceptibility | | | | 1.12 (1.06, 1.17) | 0. 9(0.81, 0.99) | 0.04 |
| Perceived severity | | | | 1.19 (1.13, 1.26) | 1.05 (0.92, 1.20) | 0.44 |
| Perceived Benefit | | | | 1.13 (1.09, 1.17) | 1.09 (1.02, 1.17) | 0.02 |
| Self-efficacy | | | | 1.02 (0.99, 1.05) | 1.01(0.96, 1.05) | 0.82 |
| Cues to action | | | | 1.07(1.04, 1.09) | 1.01 (0.96, 1.05) | 0.69 |

* P-value less than 0.05; AOR = adjusted odds ratio; COR = crude odds ratio

This is in line with the claim that knowing health practices, especially those related to HIV, will help one understand how to improve their own health and protect themselves [34].

This study also showed that participants who completed elementary school and above were significantly associated with HIV preventive practice. The findings of this study are supported by a study conducted in Gondar and 'Bahir Dar [35]. This explained that participants who had a higher level of education had knowledge of HIV preventive practice due to the fact that education leads to good preventive practice [35].

This study showed that participants who did not practice sex to earn money were positively associated with HIV preventive practice. The findings of this study are supported by a study conducted in Western Kenya [2]. This might be due to the lack of awareness regarding HIV prevention modalities among these street dwellers and the lack of attention given by the local health authority. Besides this, participants who practice sex to earn money have several factors that force them not to use preventive practices for HIV such as having many sexual partners, unsafe working conditions, physical violence including intimate partner violence, substance abuse and inconsistent condom use [36].

This study revealed that perceived susceptibility is significantly associated with the preventive practice of HIV/AIDS. A study conducted in Russia supports this finding [37]. This might be explained by the fact that if the person perceives susceptibile to HIV/AIDS, there is a high chance of using preventive practices.

Furthermore, in this study, perceived benefit is significantly associated with HIV preventive practice. This study is supported by a study conducted in Gambella town [21] and Iran [38]. This might be explained by the fact that if a person perceives that HIV preventive practice is important, there is a high chance of having HIV preventive practice.

## Strength and limitations

Participants in the study were chosen at random to create a representative sample. The chicken-egg dilemma prevents us from determining the temporal association between knowledge and HIV/AIDS preventative practice because the study was cross-sectional. It is unknown if the association is temporal because the study was cross-sectional in design.

## Conclusion

The magnitude of HIV preventive practice among street dwellers was low. Being male, practicing sex to earn money, educational status, having good knowledge about HIV preventive practice, perceived susceptibility to HIV, and perceived benefit of using HIV preventive practice were positively associated with utilization of HIV preventive practice. The purpose of this study was to assess HIV preventive practice and its associated factors among street dwellers. Therefore, responsible organizations, both governmental and non-governmental, should design inclusive strategies to improve HIV preventive practice among street dwellers by focusing on regular demand creation activities, awareness creation about HIV preventive practice, and sustainable condom distribution in the city. Future researchers should better conduct studies by using a mixed methodology and including organizations that are responsible for the health of street dwellers to enhance the utilization of HIV preventive behaviors.

## Supporting information

**S1 Text. English version questionnaire.**
(DOCX)

## Acknowledgments

The authors would like to acknowledge Bahir Dar University for providing ethical approval letter, data collectors, supervisors, and study participants for their willingness to participate in the study.

## Author Contributions

**Conceptualization:** Yosef Wassihun, Zemed Hunegnaw, Tadele Fentabel Anagaw, Zeamanuel Anteneh Yigzaw, Eyob Ketema Bogale.

**Data curation:** Yosef Wassihun, Zemed Hunegnaw, Tadele Fentabel Anagaw, Zeamanuel Anteneh Yigzaw, Eyob Ketema Bogale.

**Formal analysis:** Yosef Wassihun, Zemed Hunegnaw, Eyob Ketema Bogale.

**Investigation:** Zemed Hunegnaw, Eyob Ketema Bogale.

**Methodology:** Yosef Wassihun, Zemed Hunegnaw, Tadele Fentabel Anagaw, Zeamanuel Anteneh Yigzaw, Eyob Ketema Bogale.

**Project administration:** Eyob Ketema Bogale.

**Resources:** Zemed Hunegnaw.

**Software:** Yosef Wassihun, Zemed Hunegnaw, Tadele Fentabel Anagaw, Zeamanuel Anteneh Yigzaw, Eyob Ketema Bogale.

**Supervision:** Yosef Wassihun, Eyob Ketema Bogale.

**Validation:** Yosef Wassihun, Zemed Hunegnaw, Tadele Fentabel Anagaw, Zeamanuel Anteneh Yigzaw, Eyob Ketema Bogale.

**Visualization:** Zemed Hunegnaw, Eyob Ketema Bogale.

**Writing – original draft:** Yosef Wassihun, Zemed Hunegnaw, Tadele Fentabel Anagaw, Zeamanuel Anteneh Yigzaw, Eyob Ketema Bogale.

**Writing – review & editing:** Yosef Wassihun, Zemed Hunegnaw, Tadele Fentabel Anagaw, Zeamanuel Anteneh Yigzaw, Eyob Ketema Bogale.

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
