## [Decision Letter · Decision Letter 0]

14 Feb 2024

PGPH-D-23-02271

HIV/AIDS Preventive Practice and Its Associated Factors Among Street Dwellers in Ethiopia: Application of Health Belief Model

Dear Dr. Bogale,

Thank you for submitting your manuscript to PLOS Global Public Health. After careful consideration, we feel that it has merit but does not fully meet PLOS Global Public Health’s publication criteria as it currently stands. Therefore, we invite you to submit a revised version of the manuscript that addresses the points raised during the review process.

Please note that we have only been able to secure a single reviewer to assess your manuscript. We are issuing a decision on your manuscript at this point to prevent further delays in the evaluation of your manuscript. Please be aware that the editor who handles your revised manuscript might find it necessary to invite additional reviewers to assess this work once the revised manuscript is submitted. However, we will aim to proceed on the basis of this single review if possible. 

We look forward to receiving your revised manuscript.

Kind regards,

Vanessa Carels

Staff Editor

Journal Requirements:

Additional Editor Comments (if provided):

Reviewers' comments:

Reviewer's Responses to Questions

**Comments to the Author**

1. Does this manuscript meet PLOS Global Public Health’s publication criteria? Is the manuscript technically sound, and do the data support the conclusions? The manuscript must describe methodologically and ethically rigorous research with conclusions that are appropriately drawn based on the data presented.

Reviewer #1: Partly

2. Has the statistical analysis been performed appropriately and rigorously?

Reviewer #1: No

3. Have the authors made all data underlying the findings in their manuscript fully available (please refer to the Data Availability Statement at the start of the manuscript PDF file)?

Reviewer #1: No

4. Is the manuscript presented in an intelligible fashion and written in standard English?

Reviewer #1: Yes

5. Review Comments to the Author

Reviewer #1: "HIV/AIDS Preventive Practice and Its Associated Factors Among Street Dwellers in

Ethiopia: Application of Health Belief Model" is a good thesis which may ultimately helps to combat the disease. 

1. Street Dwellers should be given an operational definition. 

2. Line number 129-132 "than or equal to 12 years old was included. While street dwellers that were unable to communicate due to severe illness were excluded from the study.

Exclusion criteria: Street dwellers that were unable to communicate due to severe illness were excluded from the study"

First ideas were repeated? Second why those people with severe illness are there in the street and what do you do for them? What type of illness you encountered?

3. P= 50% (since there is no previous published research in Ethiopia). ? How? Preventive practice?

4. Line number 297-299 "This discrepancy might be due to differences in partners' involvement in HIV/AIDS preventive practice, demand creation activities, and sample size differences. This idea is Unclear.

5. Line number 304-307, In addition, women's use of HIV/AIDS preventative measures is impacted by a number of factors, including their greater propensity than men to sell sex in order to survive, as well as their vulnerability to sexual  exploitation, coercion, and rape. Vast generalization

6. Rewrite the conclusion

6. PLOS authors have the option to publish the peer review history of their article (what does this mean?). If published, this will include your full peer review and any attached files.

**Do you want your identity to be public for this peer review?** For information about this choice, including consent withdrawal, please see our Privacy Policy.

Reviewer #1: No

---

## [Decision Letter · Decision Letter 1]

19 Mar 2024

PGPH-D-23-02271R1

HIV/AIDS Preventive Practice and Its Associated Factors Among Street Dwellers in Ethiopia: Application of Health Belief Model

Dear Dr. Bogale,

Thank you for submitting your manuscript to PLOS Global Public Health. After careful consideration, we feel that it has merit but does not fully meet PLOS Global Public Health’s publication criteria as it currently stands. Therefore, we invite you to submit a revised version of the manuscript that addresses the points raised during the review process.

We look forward to receiving your revised manuscript.

Kind regards,

Siyan Yi, MD, MHSc, PhD

Academic Editor

Journal Requirements:

2. We have noticed that you have uploaded Supporting Information files, but you have not included a list of legends. Please add a full list of legends for your Supporting Information files after the references list.

Additional Editor Comments (if provided):

Overall, the manuscript has seen significant improvement. However, there is still a need for further refinement in writing quality, particularly in eliminating grammatical errors, typos, and the misuse of punctuation throughout the paper. I have read the abstract and provided some examples and suggestions. However, please note that these are not exhaustive. The authors need to ensure a thorough proofreading of the paper before we could accept it.

1. Follow the structure recommended by PLOSGPH, including a brief background and rationale before stating the objective.

2. Revise the objective for clarity – "To assess HIV prevention practices and associated factors among street dwellers in Bahir Dar City, Ethiopia using the Health Belief Model."

3. Lines 18-19: "A community-based cross-sectional study was conducted in Bahir Dar City from March 12 to April 30, 2023."

4. The sentence "Epi-data version 3.1 was used to enter the data, which was then exported to SPSS version for analysis" can be removed. This will provide space for more pertinent information, such as details about key variables and measurements (e.g., how 'HIV prevention practice' was defined and measured). Additionally, the data analysis description can be summarized for conciseness.

5. Summarize key participants' characteristics at the beginning of the results section.

6. Avoid using the term 'HIV/AIDS'; UNAIDS no longer recommends it. The term 'HIV/AIDS prevention' sounds awkward

7. Line 29: "Education" should not be capitalized.

8. Clarify several variables described in the abstract, such as 'good knowledge,' 'perceived susceptibility,' and 'perceived benefits.' It's important to specify what these variables refer to - knowledge, susceptibility, and benefits of what?

9. Ensure consistency throughout the paper (e.g., lines 29-32: inconsistent use of decimals in CIs).

10. In the conclusion, avoid repeating results; instead, summarize key findings supporting recommendations. Make recommendations more specific.

These revisions will enhance the clarity and coherence of the manuscript.

Reviewers' comments:

Reviewer's Responses to Questions

**Comments to the Author**

1. If the authors have adequately addressed your comments raised in a previous round of review and you feel that this manuscript is now acceptable for publication, you may indicate that here to bypass the “Comments to the Author” section, enter your conflict of interest statement in the “Confidential to Editor” section, and submit your "Accept" recommendation.

Reviewer #1: All comments have been addressed

2. Does this manuscript meet PLOS Global Public Health’s publication criteria? Is the manuscript technically sound, and do the data support the conclusions? The manuscript must describe methodologically and ethically rigorous research with conclusions that are appropriately drawn based on the data presented.

Reviewer #1: Partly

3. Has the statistical analysis been performed appropriately and rigorously?

Reviewer #1: Yes

4. Have the authors made all data underlying the findings in their manuscript fully available (please refer to the Data Availability Statement at the start of the manuscript PDF file)?

Reviewer #1: Yes

5. Is the manuscript presented in an intelligible fashion and written in standard English?

Reviewer #1: Yes

6. Review Comments to the Author

Reviewer #1: (No Response)

7. PLOS authors have the option to publish the peer review history of their article (what does this mean?). If published, this will include your full peer review and any attached files.

**Do you want your identity to be public for this peer review?** For information about this choice, including consent withdrawal, please see our Privacy Policy.

Reviewer #1: **Yes: **Yohannes Habtegirogis Abate

---

## [Editor Report · Decision Letter 2]

12 Apr 2024

HIV/AIDS Preventive Practice and Its Associated Factors Among Street Dwellers in Ethiopia: Application of Health Belief Model

PGPH-D-23-02271R2

Dear Mr Bogale,

We are pleased to inform you that your manuscript 'HIV/AIDS Preventive Practice and Its Associated Factors Among Street Dwellers in Ethiopia: Application of Health Belief Model' has been provisionally accepted for publication in PLOS Global Public Health.

Best regards,

Siyan Yi, MD, MHSc, PhD

Academic Editor